# Exercise and/or Genistein Treatment Impact Gut Microbiota and Inflammation after 12 Weeks on a High-Fat, High-Sugar Diet in C57BL/6 Mice

**DOI:** 10.3390/nu12113410

**Published:** 2020-11-06

**Authors:** Carmen P. Ortega-Santos, Layla Al-Nakkash, Corrie M. Whisner

**Affiliations:** 1College of Health Solutions, Arizona State University, Phoenix, AZ 85004, USA; cortegas@asu.edu; 2Department of Physiology, College of Graduate Studies, Midwestern University, Glendale, AZ 85308, USA; lalnak@midwestern.edu

**Keywords:** gut microbiota, genistein, exercise, inflammation, Western diet, high-fat diet

## Abstract

Genistein (Gen) and exercise (Exe) have been postulated as potential strategies to ameliorate obesity, inflammation, and gut microbiota (GM) with promising results. However, the impact of the combination of both Exe and Gen is yet to be investigated. We aimed to analyze the impacts of Exe, Gen, and their combined effects on GM and inflammation in mice after a 12-week high-fat, high-sugar diet (HFD). Eighty-three C57BL/6 mice were randomized to control, HFD, HFD + Exe, HFD + Gen, or HFD + Exe + Gen. The V4 region of the 16S rRNA gene was analyzed with Illumina MiSeq. Serum samples were used to analyze interleukin (Il)-6 and Tumor Necrosis Factor alpha (TNF-alpha). The HFD + Exe and HFD + Exe + Gen treatments resulted in significantly greater microbial richness compared to HFD. All the treatments had a significantly different impact on the GM community structure. *Ruminococcus* was significantly more abundant after the HFD + Exe + Gen treatment when compared to all the other HFD groups. Exe + Gen resulted in serum Il-6 concentrations similar to that of controls. TNF-alpha concentrations did not differ by treatment. Overall, Exe had a positive impact on microbial richness, and *Ruminococcus* might be the driving bacteria for the GM structure differences. Exe + Gen may be an effective treatment for preventing HFD-induced inflammation.

## 1. Introduction

Western diets (e.g., high-fat, high-sugar diets) have been postulated as a key factor in shifting gut microbial diversity and community structure toward dysbiosis (an imbalance in the beneficial-to-pathogenic species ratio, diversity, and abundance within the microbial community). An altered microbial structure has been associated with increased energy harvest from the diet [1] and increased inflammation [2], which both contribute to the obese phenotype [3].

Different behavioral strategies have been studied to prevent or revert the negative effects of Western diets on host health through the modification of gut microbiota [4,5,6,7]. Changes in the macronutrient composition of the diet and caloric restriction have been shown to induce short- and long-term positive changes in gut microbiota in healthy and unhealthy adults [8,9,10]. The addition of functional ingredients to habitual diets has emerged as a potential dietetic treatment to induce gut microbial changes that may elicit health benefits [11]. Among the different functional ingredients, soy isoflavones (daidzein and genistein) have exhibited the ability to minimize menopause symptoms [12], improve breast cancer outcomes [13], and maintain bone health [14]. The interindividual effects and bioavailability of isoflavones are gut microbiota composition-dependent [15]. Genistein, the most abundant isoflavone in soy, is metabolized by gut bacteria into numerous metabolites [16] that target and modulate estrogen-dependent and non-estrogenic activity pathways with a wide variety of biological activities (e.g., the reduction of inflammation) [17,18,19,20]. Recently, genistein has been highlighted to have the capacity to reduce inflammation, modify adipocyte functioning, and maintain body weight [21,22,23,24,25,26].

Exercise has been suggested as a cost-effective treatment to improve inflammation, glucose metabolism, and body weight maintenance [27]. In the last few years, there has been a growing interest in physical activity and exercise as a potential strategy for modulating the gut microbiome [28]. Animal and human trials have demonstrated promising results, whereby exercise modifies gut microbial diversity and community structure [29]. Voluntary exercise may produce a better outcome with respect to modifying gut microbiota by increasing *Bacteroidetes*. Previously, 8–12 weeks of voluntary exercise was shown to modify the gut microbiota in rodents by increasing the abundance of *Bacteroidetes* [30,31,32]. Meanwhile, 6 to 8 weeks of forced exercise in mice has been shown to increase the abundance of *Firmicutes* and enhance microbial community richness [30,33,34]. Among human trials, a cross-sectional study analyzing the impact of physical activity on menopausal women indicated a greater bacterial diversity in physically active women compared to their sedentary counterparts [35]. Exercise interventions also have an impact on the gut microbial communities of obese and lean individuals; the beta-diversity of obese individuals, after a 6-week exercise intervention, no longer differed from lean individuals [36]. Long-term exercise has also been shown to improve the gut microbiota composition in Type 2 diabetic overweight individuals by reducing the presence of pathogenic fungal species such as *Candida albicans* and *Mycete* spp. [37].

Given the preliminary results on exercise and gut microbiota research, further animal trials have tested the capacity of exercise to restore and prevent gut microbiota changes after subjection to a Western diet. Literature evidence is varied; exercise has been shown to modulate the gut microbiota towards a more diverse bacterial community, whereas sometimes exercise was not enough to restore Western diet-induced gut microbial modifications [32,38]. Isoflavones emerge as a potential treatment to prevent and improve obesity and associated metabolic effects (e.g., inflammation) through the modification of gut microbial structure. The combined effects of exercise and isoflavones on gut microbiota and their impact on obesity and inflammation have yet to be investigated. Therefore, this study aimed to analyze the impacts of either exercise or genistein, and their combined treatment on preventing obesity outcomes while mice were fed a high-fat, high-sugar diet (HFD) for 12 weeks to induce obesity. The impact of these treatments on the gut microbiota and inflammation were also assessed.

## 2. Materials and Methods

### 2.1. Mice and Exercise/Diet Protocols

Male (*n* = 41) and female (*n* = 42) C57BL/6 mice, aged 4–5 weeks, were purchased from Charles River (Wilmington, MA, USA). All procedures were approved by the Institutional Animal Care and Use Committee at Midwestern University in Glendale, AZ, USA. After 1 week of acclimatization, the mice were randomly assigned to one of five groups: control (fed standard rodent chow and water, *n* = 18), HFD (*n* = 15), HFD + exercise (HFD + Exe; *n* = 18), HFD + genistein (HFD + Gen; *n* = 18), or HFD + Exe + Gen (*n* = 14). The HFD + Exe and HFD + Exe + Gen groups performed moderate intensity exercise 5 days per week on an electrically driven treadmill (Columbus Instr., Columbus, OH, USA) for a period of 12 weeks. The training regimen consisted of a 3-week graded increase in exercise duration and intensity as follows: Week 1, 10 min at 10 m/min; Week 2, 20 min at 10 m/min; Week 3, 30 min at 12 m/min; Weeks 4–12, 30 min at 15 m/min. All mice were provided with food (Table 1) and water or sugar-supplemented water ad libitum and maintained in a room with an alternating 12 h light/dark cycle that was kept at 22 °C. The HFD (20% carbohydrate, 20% protein, and 60% fat) was purchased from Dyets (Dyets Inc, Bethlehem PA) with 32.3 g/kg of corn oil (polyunsaturated fat) and 316.6 g/kg of lard (saturated fat). The HFD + Gen and HFD + Exe + Gen groups were supplemented with 600 mg of genistein/kg of diet. The drinking water in all the groups except control was supplemented with 42 g of sugar per liter (55% fructose/45% sucrose). Body weights and general health were monitored weekly for the entire 12-week study. Animal care was conducted in accordance with established guidelines, and all protocols were approved by the Midwestern University Institutional Animal Care and Use Committee (IACUC project #2880 approved 30 October 2017).

### 2.2. Sample Collection, DNA Isolation, Preparation, and Sequencing

The fecal pellets were collected at the time of euthanasia and immediately placed in a cryotube, frozen in liquid nitrogen, and stored at −80 °C until processing. Microbial DNA was extracted and isolated from the fecal pellets using a DNeasy PowerSoil DNA Isolation Kit (12855-100, Qiagen, Hilden, Germany) according to the directions provided by the manufacturer. Bacterial community analysis was performed via next generation sequencing on the MiSeq Illumina platform (San Diego, CA, USA). Amplicon sequencing of the V4 region of the 16S rRNA gene was performed using the barcoded primer set 515f/806r [39] and following the Earth Microbiome Project (EMP) protocol (http://www.earthmicrobiome.org/emp-standard-protocols/) for library preparation. PCR amplifications for each sample were performed in triplicate, and DNA amplicons were pooled and quantified using the Quant-iT™ PicoGreen^®^ dsDNA Assay Kit (ThermoFisher, Waltham, MA, USA). A no-template control sample was included during the library preparation as a control for extraneous nucleic acid contamination. A total of 240 ng of DNA from each sample was pooled and then cleaned using the QIA quick PCR purification kit (QIAGEN, Hilden, Germany). The pooled sample was then quantified using the Illumina library Quantification Kit ABI Prism^®^ (Kapa Biosystems, Wilmington, MA, USA) and diluted to a final concentration of 4 nM. The sample was further denatured and diluted to a final concentration of 4 pM with 15% PhiX. Finally, the DNA library was loaded in the MiSeq Illumina and run using the version 2 module, with 2 × 250 paired-end reads, and following the directions of the manufacturer.

### 2.3. Serum Il-6 and TNF-Alpha

Blood was collected at the time of euthanasia by cardiac puncture, and then, serum was extracted and stored at −80 °C until use. The Il-6 and TNF-alpha concentrations were obtained via assays (Milliplex Assay Millipore, Billerica, MA, USA) using standard commercially available kits as per the manufacturers’ instructions.

### 2.4. Fecal Bile Acids

The bile acids in feces were quantified using a Waters Acquity UPLC System with a QDa single quadrupole mass detector and an autosampler (192 sample capacity). This was performed on a fee-for-service basis by Dr. E. Friedman, Microbial Culture & Metabolomics Core, PennCHOP Microbiome Program, Perelman School of Medicine, University of Pennsylvania. Briefly, fecal samples were suspended in methanol (5 μL/mg of stool), vortexed for 1 min, and centrifuged twice at 13,000× *g* for 5 min. The supernatant was transferred to a new tube, sealed, and stored at 4 °C until analysis. The samples were analyzed on an Acquity UPLC with a Cortecs UPLC C-18 + 1.6 mm 2.1 × 50 mm column. The flow rate was 0.8 mL/min, the injection volume was 4 μL, the column temperature was 30 °C, the sample temperature was 4 °C, and the run time was 4 min per sample. Eluent A was 0.1% formic acid in water, Eluent B was 0.1% formic acid in acetonitrile, the weak needle wash was 0.1% formic acid in water, the strong needle wash was 0.1% formic acid in acetonitrile, and the seal wash was 10% acetonitrile in water. The gradient was 70% Eluent A for 2.5 min, a gradient to 100% Eluent B for 0.6 min, and then 70% Eluent A for 0.9 min. Blank cells were below the limit of detection, which was at least 0.5 nmol/g of stool. The mass detection channels were +357.35 for chenodeoxycholic acid and deoxycholic acid; +359.25 for lithocholic acid; −407.5 for cholic, alphamuricholic, betamuricholic, gamma muricholic, and omegamuricholic acids; −432.5 for glycolithocholic acid; −448.5 for glycochenodeoxycholic and glycodeoxycholic acids; −464.5 for glycocholic acid; −482.5 for taurolithocholic acid; −498.5 for taurochenodeoxycholic and taurodeoxycholic acids; and −514.4 for taurocholic acid. The samples were quantified against standard curves of at least five points run in triplicate. Standard curves were run at the beginning and end of each metabolomics run. Quality control checks (blanks and standards) were run every eight samples. The range of the assay was at least 50–10,000 nM, the limit of detection was <50 nM, and the limit of quantitation was >10,000 nM.

### 2.5. Sequence Analysis and Statistics

Taxonomic analysis [2,3,4,5,6,7], bacterial diversity (alpha and beta diversity) analysis, and visualization of the data were performed using Quantitative Insights into Microbial Ecology (QIIME2), version 2019.1 [40]. First, using the Divisive Amplicon Denoising Algorithm 2 (DADA2) pipeline, the de-multiplexed sequence files were filtered and trimmed by trimming the forward reads at 251 bp and the reverse reads at 0 bp after the visual inspection of quality plots (quality score > 20–25). Next, the forward and reverse reads were merged into one unique sequence file per sample, and chimeric sequences and other errors formed during PCR amplification were removed. Lastly, the inferred samples were combined into one unified sequence table. The DADA2 analysis output was used to choose a rarefaction depth of 15,247 counts per sample, keeping all the sequenced samples for final statistical analyses. The analysis of α-diversity was assessed using the Shannon Diversity index [40,41,42,43,44,45,46,47,48,49], Observed Operational Taxonomic Units (OTUs) [40,41,42,43,44,45,46,47,48], Faith’s Phylogenetic Diversity (PD) [40,41,42,43,44,45,46,47,48,49], and Pielou’s evenness index [40,41,42,43,44,45,46,47,48] metrics. Group differences in the alpha diversity metrics were analyzed using Kruskal–Wallis nonparametric tests. For β-diversity, the Jaccard [40,42,43,45,46,47,48,50,51], Bray–Curtis [40,42,43,45,46,47,48,50,51], and weighted [5,6,7,9,10,11,12,13,16,17,18,19,20] and unweighted UniFrac [5,6,7,9,10,11,12,13,16,17,18,19,20] distance metrics were generated from QIIME2 and visualized using EMPeror. Group differences in the beta-diversity metrics were assessed by permutational multivariate analysis of variance (PERMANOVA) [51]. The comparison of the differential abundance of microbial taxa among the five groups was performed using the Linear discriminate analysis of effect size (LEfSe) [52] tool (Galaxy version 1.0; http://huttenhower.sph.harvard.edu/galaxy). Differential abundance was calculated by a non-parametric Kruskal–Wallis test and consequent pairwise Wilcoxon rank-sum testing to detect which taxa were differentially abundant with an alpha value of 0.05 and Linear Discriminant Analysis (LDA) score > 2 [53]. Alpha was set at 0.05 for all tests of significance, and the Benjamini–Hochberg correction (adjusted p-values) was used for multiple comparisons following PERMANOVA analysis.

In addition, the serum levels of Il-6 and TNF-alpha were prepared for group comparisons using SPSS version 25. The normality of the data was assessed with the Shapiro–Wilk statistical test. Because the animal body weight (BW) was not normally distributed, the baseline BW, post-intervention BW, and BW change data were assessed using the Kruskal–Wallis non-parametric test followed by a post-hoc Dunn test and correction for multiple comparisons using the Benjamini–Hochberg method. A general linear model (GLM) was used to analyze the effect of study treatment (independent variable) on the serum concentrations of Il-6 and TNF-alpha (dependent variables) while adjusting for animal sex as a covariate. Using the Bonferroni post-hoc test, significant interactions and main effects for all statistical tests were analyzed. Differences were considered significant when p-values were <0.05.

## 3. Results

### 3.1. Descriptive and Body Weight Change

The descriptive characteristics of the animals are described in Table 2. There was a statistical difference (*p* < 0.05) in body weight change among the groups. The change in body weight (WT) after 12 weeks of intervention was significantly lower in the control group treatments. The group receiving exercise and genistein had a significantly lower body weight change compared to the HFD + Exe, HFD + Gen, and HFD groups. These results suggest that the combination of both treatments may have a preventive synergetic effect on WT gain when consuming an HFD (Table 2). The food and water intake of the animals is described in Table 3.

### 3.2. Inflammation

After adjusting for animal sex, we found a statistically significantly (*p* < 0.05) higher Il-6 concentration in the HFD group (96.04 ± 14.02 pg/mL) compared to controls (27.57 ± 15.62 pg/mL). The Exe and Gen treatments prevented the pro-inflammatory effects of the HFD, as no significant differences (*p* > 0.05) were found when compared to the control group. As expected, the combined treatment (HFD + Exe + Gen) resulted in lower Il-6 concentrations (52.09 ± 28.14 pg/mL), followed by the HFD + Exe (58.16 ± 41.76 pg/mL), and HFD + Gen (62.44 ± 45.89 pg/mL) treatments, but these differences were not statistically significant. We did not find any significant differences between groups after adjusting for sex for the TNF-alpha concentration.

### 3.3. Alpha Diversity

The Observed OTUs (Figure 1), a quantitative measure of community richness, were statistically significantly higher in the control group compared to the HFD (Kruskal–Wallis H = 12.21, *p* < 0.01) and HFD + Gen (Kruskal–Wallis H = 10.89, *p* < 0.01) treatments. This suggests that HFD reduces community richness compared to a standard diet. Exercise also had an impact on the community richness, as the HFD + Exe and HFD + Exe + Gen groups had significantly higher Observed OTUs than the HFD + Gen (Kruskal–Wallis H = 9.954, *p* < 0.01; Kruskal–Wallis H = 10.31, *p* < 0.01, respectively) and HFD-fed animals (Kruskal Wallis H = 12.71, *p* < 0.001; Kruskal Wallis H = 9.78, *p* < 0.001, respectively). Overall, these results suggest that consuming a HFD reduces community richness and supplementing the HFD with Exe prevents the deleterious effects of the high-fat, high-sugar diet on community richness, as evidenced by no significant differences between the exercise groups (HFD + Exe and HFD + Exe + Gen) and control group. When the HFD was supplemented with Gen and Exe, the community richness was higher compared to the HFD + Gen group; however, no difference was observed between the combined HFD + Exe + Gen and HFD + Exe groups, suggesting that exercise may have a greater impact on gut microbial community richness than genistein.

Faith’s PD (Figure 2), a qualitative measure of community richness that incorporates the phylogenetic relationships between the features, was statistically significantly higher in the control group when compared to HFD (Kruskal–Wallis H = 10.18, *p* < 0.001) and HFD + Gen (Kruskal–Wallis H = 11.33, *p* < 0.01) groups. There was significantly greater bacterial richness (Faith’s PD) in the HFD + Exe and HFD + Exe + Gen groups when compared to HFD (Kruskal–Wallis H = 12.04, *p* < 0.01; Kruskal–Wallis H = 9.02, *p* < 0.01, respectively). The HFD + Exe + Gen treatment resulted in greater richness compared to HFD + Gen (Kruskal–Wallis H = 12.63, *p* < 0.01). A comparison of the Pielou’s Evenness index data revealed no differences (*p* > 0.05) among the groups. Similarly, the Shannon Diversity Index, which accounts for both richness and evenness, did not differ significantly between any of the treatment groups (*p* > 0.05). The relative frequencies of taxa at the phylum level (Appendix A) were calculated per group.

### 3.4. β-Diversity

The β-diversity index results according to the Jaccard distance, a metric that compares community similarities based on microbial presence/absence, were statistically different between all the study treatment groups (Figure 3), suggesting that each condition exerted a different effect on the community structure. The results from Bray–Curtis (Figure 4), a metric that accounts for differences in microbial community dissimilarities based on microbial abundance and the presence/absence of microbial features, also suggested that all the treatments resulted in significantly (*p* < 0.05) different microbial community structures. These results suggest that the combination of Exe and Gen had a unique impact on the microbial community structure compared to the HFD + Exe and HFD + Gen treatments. The unweighted UniFrac distance metric, a qualitative measure of community dissimilarities that considers phylogenetic relationships between and the presence/absence of microbial features, was significantly different (*p* < 0.05) among all of the groups (Figure 5). The weighted UniFrac distance metric, a quantitative measure of community dissimilarities considering the phylogenetic relationships, presence/absence of unique taxa, and abundance of microbial features, revealed that the control group statistically differed (*p* < 0.05) from HFD (*pseudo F* = 3.79, *p* = 0.016), HFD + Exe (*pseudo F* = 4.785, *p* = 0.004), HFD + Gen (*pseudo F* = 4.35, *p* = 0.001), and HFD + Exe + Gen (*pseudo F* = 7.34, *p* = 0.001). The analysis also revealed that HFD had an impact on the community structure, since there were statistically significant differences when compared to the HFD + Exe + Gen (*pseudo F* = 7.91, *p* = 0.001) group, and trends when compared to the HFD + Exe (*pseudo F* = 2.32, *p* = 0.08) and HFD + Gen (*pseudo F* = 2.36, *p* = 0.078) groups. The unique effects of the Exe, Gen, and Exe + Gen treatments remained significantly different (*p* < 0.05) when considering the phylogenetic relationships among the taxa (Figure 6). These results suggest that combining Exe and Gen has a different impact on the gut microbial structure when compared to each of the treatments alone.

### 3.5. Linear Discriminate Analysis of Effect Size (LEfSe)

All-against-all LEfSe analysis was utilized to identify the genera that most likely explained the differences between all of the treatment groups. The LEfSe results suggested that Gram-positive *Ruminococcus* were differentially more abundant in the control group when compared to all other treatment groups (Linear Discriminant Analysis (LDA) score > 2, *p* < 0.05). In order to assess the impact of genistein, exercise, and combined (Exe + Gen) treatments on gut microbiota abundance, we compared HFD vs. HFD + Exe, HFD + Gen, and HFD + Exe + Gen treatments separately. The HFD + Exe + Gen group resulted in a significantly greater abundance of *Ruminococcus* when compared to all other HFD groups (Linear Discriminant Analysis (LDA) score > 2, *p* < 0.05). When using sex as a covariate with the subclass option in LEfSe, we did not find any significant differences among the groups (*p* > 0.05).

### 3.6. Fecal Bile Acids

We found an increase in deoxycholic acid, which has previously been shown to correlate with increased intestinal carcinogenesis [54], in the HFD male group, whereas the HFD female group did not experience a significant increase compared to the other groups (Figure 7D). We found that the bile salt, secondary bile acid (SBA), and total fecal bile acid (BA) concentrations were significantly increased in the HFD + Exe + Gen group compared to the HFD group (Figure 7A–C). These results may indicate that the impact of exercise + genistein on microbial richness may positively impact the BA concentration.

## 4. Discussion

This study examined the impacts of exercise, dietary supplementation with genistein (600 mg of genistein/kg of diet), and their combined effects on the gut microbial community structure in male and female adult C57BL/6 mice consuming a HFD for 12 weeks. While the control group, fed a standard diet, had a more diverse gut microbiota compared to all of the other experimental groups, we found that exercise exerted significant impacts on gut microbiota richness (Observed OTUs and Faith’s PD index). The exercise groups (HFD + Exe and HFD + Exe + Gen) had greater alpha diversity when compared with the HFD + Gen group. Exercise appeared to prevent the deleterious effects of a HFD on bacterial diversity as evidenced by a non-significant difference in bacterial richness between the control and exercise groups (HFD + Exe and HFD + Exe + Gen). The structure of the bacterial communities differed among all five groups (Jaccard distance, Bray–Curtis dissimilarities, weighted Unifrac, and unweighted Unifrac), indicating unique effects of each treatment on the gut microbiota structure. LEfSe revealed a greater abundance of the genus *Ruminococcus* in the control group when compared to all of the other groups and in the HFD + Exe + Gen group when compared to all other HFD groups. Exe (HFD + Exe and HFD + Exe + Gen) may have a stronger influence on the BA concentrations, as evidenced by similar concentrations relative to the control group. However, significantly higher concentrations of fecal BA and SBA were observed compared to the HFD and HFD + Gen groups. The incorporation of Exe, Gen, or their combined treatment appeared to improve the inflammation (Il-6) induced by the HFD, as these intervention treatments resulted in inflammation similar to that in the control animals.

Genistein has emerged as a potential therapy for obesity [55] with beneficial effects on adipose tissue, glucose metabolism, and gut microbiota across a wide range of doses (2–600 mg/kg of body weight) in animal studies [56,57,58,59] and in human clinical trials (50–500 mg/kg of body weight) [55]. Contrary to previous findings, the supplementation of a HFD with 600 mg/kg of diet of genistein did not prevent the modification of gut microbiota alpha diversity (Shannon index, Observed OTUs, Faith’s PD, and Pielou’s evenness) and relative abundance, relative to control treatment. There is evidence of 24 weeks on a HFD supplemented with genistein (3 mg/kg of body weight/day) being significantly effective at increasing microbial alpha diversity (Observed OTUs and Shannon index) compared to a HFD alone in C57BL/6J mice [26,58]. In terms of community structure, we found beta-diversity differences among the treatment groups, which are similar to other findings [60]. Our results suggest a distinctive effect of genistein on microbial community structure; further research should aim to replicate our results and investigate what consequences the unique changes induced by genistein have at a systemic level (e.g., gut microbial metabolites in the bloodstream) in obese animals or individuals.

Beneficial effects of exercise on gut microbiota have been shown in various animal studies with HFD feeding [32,38,61,62]. In contrast to the genistein-only treatment in our study, exercise (HFD + Exe) showed a trend of preventing the HFD-induced decline in gut microbial richness (Observed OTUs and Faith’s PD); however, when measuring the relative abundance and evenness of the species present (Shannon Diversity index), the gut microbiota diversity difference disappeared. A longer trial could help to elucidate the richness trend found in this study and further investigate the impact of Exe on microbial abundance. Our data suggest that exercise may impact the types of species present but not how evenly the species are dispersed in the community. Ribeiro et al. showed that C57BL/6J male mice fed a HFD with 8 weeks of low-to-moderate (50% VO_2max_) aerobic forced exercise (30 min/day, 5 days/week) did not experience modifications in gut microbial diversity and community structure [38]. Other studies have shown that 6 weeks of high-intensity interval training (HIIT) reverses the reduction of alpha diversity and improves the Bacteroides/Firmicutes ratio after high-fat feeding in eight-week-old male C57BL/6J mice [61]. The intensity (low-to-moderate vs. high-intensity) and mode of the exercise (voluntary vs. forced) [34] training likely explains the different results among the literature [63]. Our results add to the existing literature reporting that forced exercise may be a more effective treatment to modify diversity compared to voluntary exercise [10,31,34]. Exercise resulted in a distinctive microbial community (beta diversity) compared to all groups. This suggests that exercise had a distinct impact on gut bacterial diversity (richness and community structure), as discussed elsewhere, yet it is to be discovered if it is a positive [30,32,33] or neutral effect [64]. Future interventions should test different exercise intensities to clarify the ones with major therapeutic effects to revert HFD-induced deleterious changes in gut microbiota.

The combined treatment of isoflavones and exercise is a treatment widely used to reduce menopausal symptoms and its physiological consequences (e.g., increased LDL cholesterol) [65,66]. The potential combination of both treatments to enhance gut microbiota diversity in obesity has only been studied once. Cross et al. [60] found that low-running-capacity ovariectomized rats fed with a soy-rich diet (590 mg/kg diet of isoflavones; 190 mg of genistein/kg of diet) for 28 weeks experienced a reduction in species richness (Observed OTUs, Chao1 index) compared to standard chow-fed rats. Similarly, we found that exercise and genistein impacted the gut microbial community differently than either treatment alone. The addition of genistein was not as effective as exercise, as evidenced by the differences between the HFD + Exe and HFD + Exe + Gen groups, which were clustered closer in the PCoA plots. This suggests that exercise may have a stronger impact on community structure compared to genistein alone, which clustered further apart from the exercise groups.

We found interesting results when we analyzed the taxonomy driving the differences between groups with LEfSe. The combined treatment group (HFD + Exe + Gen) was dominated by the genus *Ruminococcus*, when compared to all of the other HFD groups. When we compared all groups against each other, *Ruminoccocus* dominated the control group, suggesting that the combined treatment may drive taxon dominance toward a healthy control microbiota as found by others [13,58]. The increased abundance of *Ruminococcus* spp. could be related to its involvement in the conversion of the isoflavones from the glycosides to the aglycone forms, the bioactive metabolites of isoflavones [67]. Our results indicated that the combination of exercise and genistein compared to other intervention groups was significantly driven by *Ruminococcus* spp. The combined lifestyle approach of consuming genistein and exercise may have an additive effect to prevent HFD-induced gut microbial disturbances by modifying gut microbial structure and richness. However, more research is required to confirm or reject those effects, and the potential systemic effects (e.g., on inflammation or the BA pool) of these changes are undetermined. Further analysis should include the concentrations of metabolites such as aglycones and 5-hydroxy-equol in the gut and blood samples to measure the synergistic effect of genistein and exercise on the conversion of bioactive isoflavone metabolites prior to and after the intervention.

Contrary to the literature [68,69], we did not find any significant differences among the groups after adjusting for sex based on the phenotype characteristics. Contrary to our findings, Hwang et al. found that, despite C57BL/6J mice having a similar energy intake, males experienced significantly greater increases in body weight compared to females [68]. Female estrogen may provide enough protection against the metabolic effects and risk of increased body weight [70,71]. The amount of food intake by the HFD groups may explain our results. However, future research should continue using sex as a biological variable to further understand how sex influences phenotypic and cardiometabolic factors in a diet-induced obesity model.

As expected, we found that 12 weeks of a HFD led to a significant increase in the pro-inflammatory cytokine Il-6 when compared to all other treatment groups. In line with the literature, genistein [6,72,73,74], exercise [75,76,77], and their combined treatment [78] showed the capacity to prevent the pro-inflammatory effects of a HFD, as inflammation after 12 weeks of treatment was similar to that of the control group. We found no significant differences when assessing TNF-alpha among groups adjusting for sex. In in vitro and human studies in different diseases, Gen has indicated the capacity to significantly reduce Il-6 and other inflammatory biomarkers (e.g., TNF-alpha) [79,80] but also no effects in other studies [81,82]. The combination of exercise with functional ingredients may have a greater impact due to a potential synergistic effect on inflammatory markers. We showed a trend for the combination of genistein and exercise in effects on inflammation, which is similar to the effects of other functional ingredients (e.g., resveratrol) in conjunction with exercise on preventing the deleterious effects of a HFD [78,83]. The literature hypothesizes that the different impacts of genistein and exercise on systemic inflammation might be driven by differences in the metabolism of the isoflavones by gut microbiota [28,84].

Similar to the inflammation results, the total fecal BAs, SBAs, and bile salts were significantly impacted by exercise and the combination of exercise and genistein. The gut microbiota composition determines the total BA pool, and vice versa [85]. Some of the gut bacteria (e.g., *Firmicutes*) are able to metabolize the BA to SBA, while changes in the BA pool could lead to alterations of the bacterial community (e.g., a contribution to the survival of bacteria involved in BA metabolism) [86]. The increased gut bacterial richness found in the Exe groups may be related to the significantly increased concentrations of total BA, SBA, and bile salts shown in the same groups (HFD + Exe and HFD + Exe + Gen) in contrast to the rest of the groups. Besides not finding any significant differences in BA concentration in the HFD + Gen vs. HFD groups, other literature found that flavonoids positively impacted the BA pool [87,88]. Less is known about the effects of Exe on fecal BA [89,90,91]. Similar to our results, Hagio et al. [90] showed a 4-week voluntary wheel-running exercise with different-sugar (dextrin, sucrose, and lactose)-supplemented diets increased fecal BA compared to the control group in rats. Others have found that 2 weeks of voluntary exercise running in male mice significantly increased the total BA fecal concentration; however, they were not under any specific dietary plan, as in not a HFD, which does not allow us to compare the combined effect of exercise and diet [92]. In humans, the Exe effect on the BA pool is not clear, and the literature is scarce [91]. Interestingly, *Ruminococcus* spp. *(Clostrium cluster XIVa/XIV)*, one of the bacteria involved in BA hydrolysis to SBA in the colon [93], was significantly increased in the HFD + Exe + Gen group compared to the rest of the HFD groups. The increase in total fecal BA may be related to the increased abundance of *Ruminoccocus* spp. from the combination of Exe and Gen shown in our study. However, further research is required to replicate our results and show the potential to enhance the action of combining exercise and isoflavones for modifying gut microbiota and, consequently, the BA pool.

The use of combined lifestyle approaches (e.g., isoflavones and exercise) to prevent HFD-induced gut microbial and inflammatory changes is an emerging field in obesity research. The strengths of this study include that it was a well-controlled investigation of diet and exercise, it utilized a large sample size (*n* = 83), and both sexes were represented. Although rodent models are often criticized for being poor predictors of human reactions, the amount of genistein used (600 mg/kg of diet; ~4–7 μM serum genistein) [94] was comparable to physiological doses when humans consuming soy milk (~2–4 μM serum genistein) [95], thereby facilitating the translation of our results to future human trials. The limitations of the present study were the absence of variables to analyze the metabolomics and the molecular mechanism behind the differences observed in this study, where exercise and genistein drove different changes in the gut microbiota structure after 12 weeks of a HFD. The absence of baseline fecal pellets did not allow us to assess differences within groups before and after the intervention.

## 5. Conclusions

Overall, we present preliminary results for an under-explored field for preventing HFD-induced inflammation with a combination of genistein and exercise. Our results suggest that both exercise and genistein had a positive impact on shifting the gut microbiota and total BA; however, exercise treatments may have a greater capacity to change alpha diversity. At the same time, the combination of exercise and genistein could prevent gut microbial community structure changes, specifically by increasing the abundance of *Ruminococcus*, the dominant microbial taxon in the healthy control animals. The increase in *Ruminococcus* could explain the similar BA profiles (increased concentrations of total BA, SBA, and bile salts) between the control and HFD + Exe + Gen groups. Further research must be performed on the combination of genistein and exercise to elucidate the microbial mechanisms and related systemic health implications.

## Figures and Tables

**Figure 1 nutrients-12-03410-f001:**
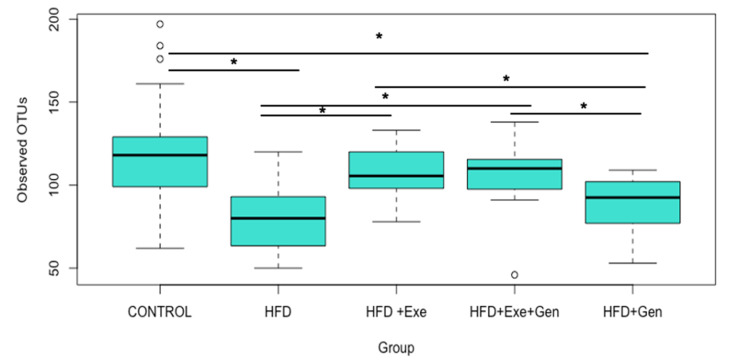
Microbial community richness (Observed Operational Taxonomic Units (OTUs)) comparisons in C57BL/6 mice (*n* = 83) after treatment with control; high-fat, high-sugar diet (HFD); HFD + Exercise (HFD + Exe); HFD + Genistein (HFD + Gen); and HFD + Exe + Gen. * denotes significant group differences according to Kruskal–Wallis (*p* < 0.05) tests after Benjamini–Hochberg correction for multiple comparisons.

**Figure 2 nutrients-12-03410-f002:**
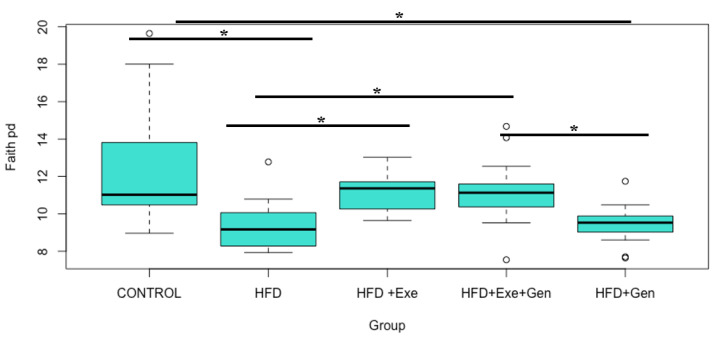
Microbial community richness (Faith’s Phylogenetic Diversity) comparisons in C57BL/6 mice (*n* = 83) after treatment with control; high-fat, high-sugar diet (HFD); HFD + Exercise (HFD + Exe); HFD + Genistein (HFD + Gen); and HFD + Exe + Gen. * denotes significant group differences according to Kruskal–Wallis test (*p* < 0.05) after Benjamini–Hochberg correction for multiple comparisons.

**Figure 3 nutrients-12-03410-f003:**
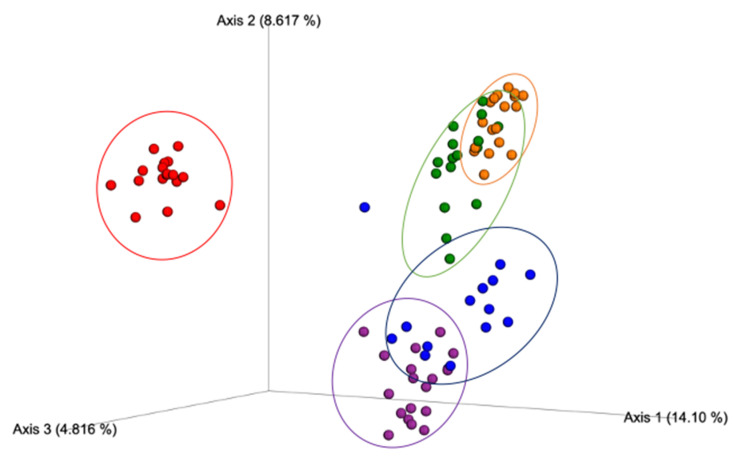
Beta-diversity PCoA plot of between-group distances using Jaccard distance matrix values. Treatment groups are as follows: control, red; high-fat, high-sugar diet (HFD), blue; HFD + genistein (Gen), purple; HFD + exercise (Exe), orange; HFD + Exe + Gen, green. Significant group differences (*p* < 0.05) after Benjamini–Hochberg correction for multiple comparisons were observed for all pairwise treatment comparisons.

**Figure 4 nutrients-12-03410-f004:**
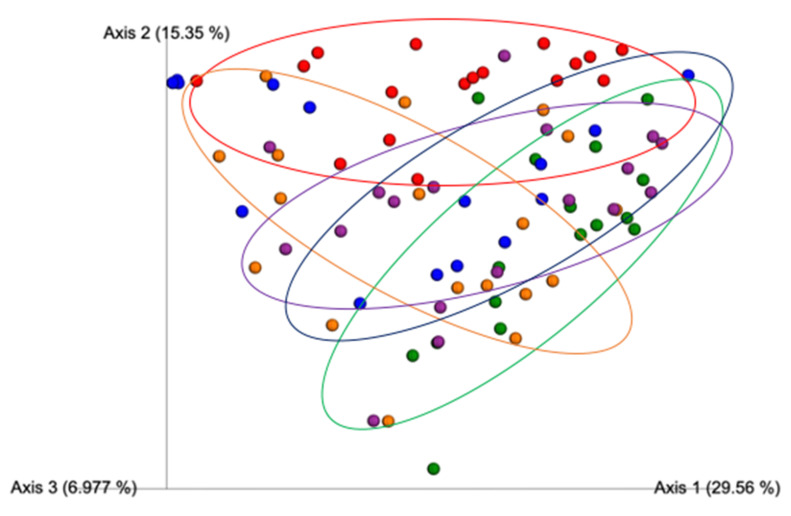
Beta diversity PCoA plot of between-group distances using Bray–Curtis distance matrix values. Treatment groups are as follows: control, red; high-fat, high-sugar diet (HFD), blue; HFD + genistein (Gen), purple; HFD + exercise (Exe), orange; HFD + Exe + Gen, green. Significant group differences (*p* < 0.05) after Benjamini–Hochberg correction for multiple comparisons were observed for all pairwise treatment comparisons.

**Figure 5 nutrients-12-03410-f005:**
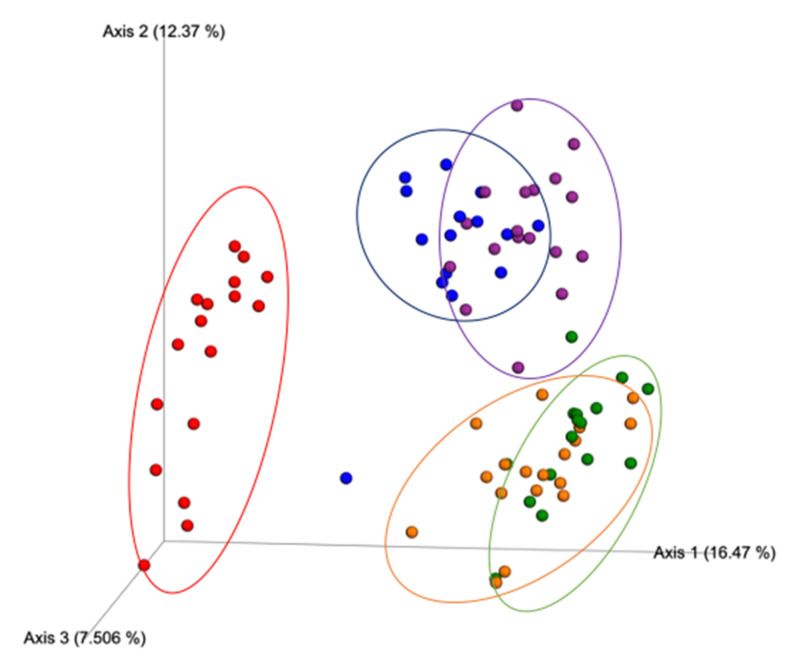
Beta diversity PCoA plot of between-group distances using unweighted UniFrac distance matrix values. Treatment groups are as follows: control, red; high-fat, high-sugar diet (HFD), blue; HFD + genistein (Gen), purple; HFD + exercise (Exe), orange; HFD + Exe + Gen, green. Significant group differences (*p* < 0.05) after Benjamini–Hochberg correction for multiple comparisons were observed for all pairwise treatment comparisons.

**Figure 6 nutrients-12-03410-f006:**
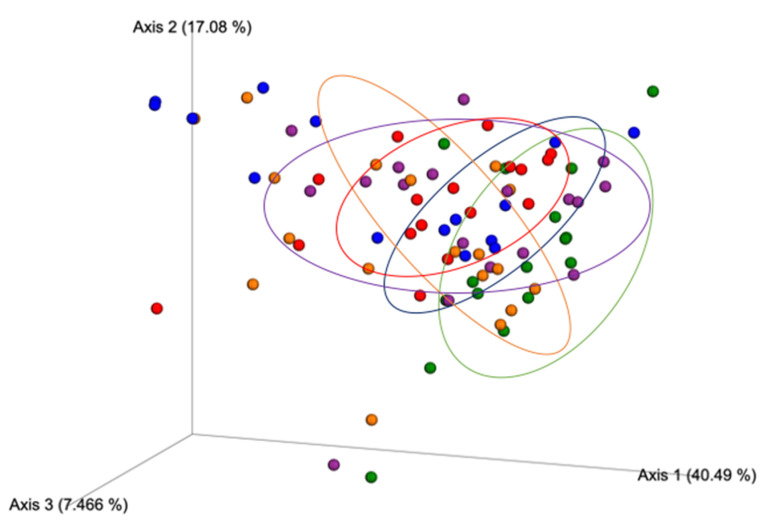
Beta diversity PCoA plot of between-group distances using weighted UniFrac distance matrix values. Treatment groups are as follows: control, red; high-fat, high-sugar diet (HFD), blue; HFD + genistein (Gen), purple; HFD + exercise (Exe), orange; HFD + Exe + Gen, green. Significant group differences (*p* < 0.05) after Benjamini–Hochberg correction for multiple comparisons were observed for all pairwise treatment comparisons.

**Figure 7 nutrients-12-03410-f007:**
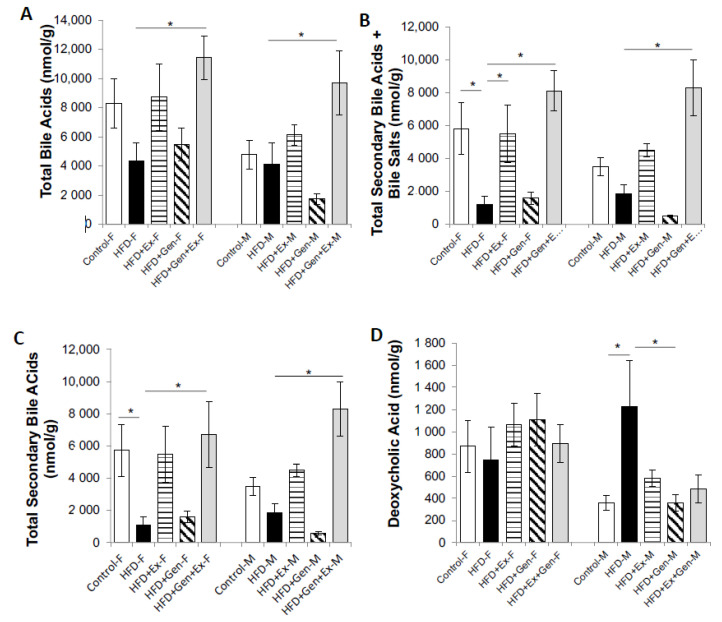
Fecal (**A**) total bile acid concentration (nmol/g); (**B**) total secondary bile acid + bile salt concentration (nmol/g); (**C**) total secondary bile acids (nmol/g); (**D**) deoxycholic acid concentration (nmol/g) in feces. Significant group differences (*p* < 0.05) were assessed by one-way ANOVA with multiple Dunnett’s comparisons. * denotes significant difference (*p* < 0.05). HFD, high-fat, high-sugar diet; Exe, exercise; Gen, genistein; F, female; M, male.

**Table 1 nutrients-12-03410-t001:** Dietary composition.

Ingredients	Chow Diet	High-Fat Diet
Carbohydrates, %	58	20
Protein, %	28.5	20
Fat, %	13.5	60
Genistein, mg/kg diet	200	600

**Table 2 nutrients-12-03410-t002:** Phenotype characteristics.

	Group				
	Control	HFD	HFD + Exe	HFD + Gen	HFD + Exe + Gen
Male, *n*	8	8	9	8	6
Female, *n*	9	8	9	9	9
Baseline body weight (WT), g ^§^	20.55 (18.21, 22.53)	20.1 (17.84, 22.96)	19.98 (17.73, 21.44)	19.78 (18.39, 23.23)	19.41 (17.10, 20.8)
Post intervention body WT, g ^§^	28.21 (24.05, 37.67)	42.76 (32.51, 49.15)	38.52 (32.02, 44.41)	37.32 (27.97, 43.62)	31.07 (25.91, 36.98)
Change in body WT, g ^§^	9.21 (4.83–14.80)	22.67 (12.29–28.71) *	18.88 (13.02–23.29) *	14.09 (9.28–18.77) *	7.72 (6.8–11.25) ^¥^
^a^ Heart WT, g	0.146 ± 0.26	0.162 ± 0.29	0.160 ± 0.03	0.140 ± 0.02	0.136 ± 0.02
^a^ Liver WT, g	1.157 ± 0.30	1.909 ± 0.90	1.430 ± 0.45	1.094 ± 0.26	0.959 ± 0.22
^a^ Kidney WT, g	0.342 ± 0.07	0.400 ± 0.11	0.394 ± 0.10	0.324 ± 0.5	0.319 ± 0.06
^a^ Adipose tissue WT, g	1.50 ± 1.23	3.657 ± 1.68	3.571 ± 1.53	3.936 ± 3.98	2.027 ± 2.09
^b,c^ (Il-6), pg/mL	27.51 ± 15.62	96.04 ± 14.02 *	58.16 ± 41.76	62.44 ± 45.89	52.09 ± 28.14
^b,c^ (TNA-alpha), pg/mL	3.72 ± 1.34	5.07 ± 1.70	3.72 ± 2.00	4.12 ± 1.63	4.86 ± 1.73

All values are mean ± standard deviations, or median (IQR) ^§^. ^a^ The weight values for heart, liver, kidney, and adipose tissue are postmortem. ^b^ The serum concentrations for Il-6 and TNF-alpha were measured after the 12-week intervention. ^c^ The values of Il-6 and TNF-alpha were based on 7–8 mice per group. WT, weight. * denotes significant differences (*p* < 0.05) compared to control group; ^¥^ denotes significant differences (*p* < 0.05) compared to HFD, HFD + Exe, and HFD + Gen. HFD: high-fat, high-sugar diet; Exe: exercise; Gen: Genistein; WT: weight.

**Table 3 nutrients-12-03410-t003:** Dietary and water intake.

	Group				
	Control	HFD	HFD + Exe	HFD + Gen	HFD + Exe + Gen
Dietary intake, g	3.88 ± 0.39	5.45 ± 2.33	8.78 ± 8.11	4.20 ± 1.88	7.79 ± 4.20
Water intake, g	2.78 ± 0.83	3.255 ± 0.85	3.90 ± 1.25	3.54 ± 0.58	4.11 ± 0.37

All values are mean ± standard deviations.

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
