# Peer review of "Exercise and/or Genistein Treatment Impact Gut Microbiota and Inflammation after 12 Weeks on a High-Fat, High-Sugar Diet in C57BL/6 Mice"

_nutrients, 2020, doi:10.3390/nu12113410_

Round 1

Reviewer 1 Report

The study by Ortega-Santos et al. studied the impact of exercise and/or Genistein on high-fat diet in mice model. There are several issues with this study, some of which should be easy to address but others more profound.

Major issues:

Material and methods

  • In line 91, the sugar content in drinking water was not described clearly. A 42 g of sugar in how much drinking water?
  • Line 117 and 284. Titled Serum Bile Acids, but only bile acids in feces were mentioned in the context. And how the sample were prepared or treated before UPLC analysis was not described.

Results:

  • Please provide how much diet and water were taken of each group during the study.
  • Since the microbiome is the principle factor to be studied, the OTU table of each group should be provided as supplementary materials.

Minor:

  • Line 324, Observer OTUs should be Observed OTUs.

Reviewer 2 Report

The manuscript title “Exercise and/or Genistein treatment impact gut microbiota and inflammation after 12 weeks on a high-fat, high-sugar diet in C57BL/6 mice” needs some improvement. These are the comments:

Major Comments:

  1. The changes in blood parameters like cholesterol, high-density lipoprotein (HDL), low-density lipoprotein (LDL) and leptin should be included to interpret the effect of exercise and Genistein treatment on high fat diet induced weight gain.
  2. Changes in relative frequencies of the most common phyla like Bacteriodes, Actinobacteria, Firmicutes and Proteobacteria should be included in gut microbial analysis as these play a significant role in gut homeostasis and high fat diet induced gut inflammation.
  3. To conclude inflammatory effect only IL-6 cytokine is not sufficient to draw the conclusion. Other inflammatory markers like TNF-α and IL-1β should also be analyzed supported by HE staining to conclude the results.
  4. How the confounding factor, gender, is affecting this study should be addressed in detail in the analysis. As the male is more vulnerable to high fat induced obesity and induced resistance.

Minor comments:

  1. There is a discrepancy in the weight gain data in text vs table 2 (especially for HFD+Gen & HFD + Exe + Gen).
  2. Line 114, says serum was collected by cardiac puncture, it is better to write blood was collected and serum was extracted further.
